# Negative Photoconductivity in 2D α-MoO_3_/Ir Self-Powered Photodetector: Impact of Post-Annealing

**DOI:** 10.3390/ma16206756

**Published:** 2023-10-19

**Authors:** Mohamed A. Basyooni-M. Kabatas, Shrouk E. Zaki, Khalid Rahmani, Redouane En-nadir, Yasin Ramazan Eker

**Affiliations:** 1Department of Precision and Microsystems Engineering, Delft University of Technology, Mekelweg 2, 2628 CD Delft, The Netherlands; 2Department of Nanotechnology and Advanced Materials, Graduate School of Applied and Natural Science, Selçuk University, Konya 42030, Turkey; 3Department of Physics, Ecole Normale Supérieure (ENS), Mohammed V University, Rabat 10140, Morocco; 4Laboratory of Solid-State Physics, Faculty of Sciences Dhar el Mahraz, University Sidi Mohammed Ben Abdellah, P.O. Box 1796, Atlas Fez 30000, Morocco; 5Department of Basic Sciences, Faculty of Engineering, Necmettin Erbakan University, Konya 42090, Turkey; yeker@erbakan.edu.tr; 6Science and Technology Research and Application Center (BITAM), Necmettin Erbakan University, Konya 42090, Turkey

**Keywords:** negative photoconductivity, thin film, photodetector, plasmonic, 2D oxide semiconductors

## Abstract

Surface plasmon technology is regarded as having significant potential for the enhancement of the performance of 2D oxide semiconductors, especially in terms of improving the light absorption of 2D MoO_3_ photodetectors. An ultrathin MoO_3_/Ir/SiO_2_/Si heterojunction Schottky self-powered photodetector is introduced here to showcase positive photoconductivity. In wafer-scale production, the initial un-annealed Mo/2 nm Ir/SiO_2_/Si sample displays a sheet carrier concentration of 5.76 × 10^11^/cm², which subsequently increases to 6.74 × 10^12^/cm² after annealing treatment, showing a negative photoconductivity behavior at a 0 V bias voltage. This suggests that annealing enhances the diffusion of Ir into the MoO_3_ layer, resulting in an increased phonon scattering probability and, consequently, an extension of the negative photoconductivity behavior. This underscores the significance of negative photoconductive devices in the realm of optoelectronic applications.

## 1. Introduction

The conversion of light into electrical energy typically relies on two fundamental mechanisms: positive photoconductivity (PPC) and negative photoconductivity (NPC). In traditional PPC-based photodetectors, the generation of photocurrent occurs when incident photons possess energy levels greater than the semiconductor material’s band gap (ΔE_g_) [1,2]. Consequently, PPC-based photodetectors can effectively detect photons with wavelengths of λ that are equal to or less than ΔE_g_/hc, where h represents Planck’s constant. Nevertheless, PPC photodetectors exhibit several inherent limitations, including narrow optical bandwidths, high dark currents, restricted operation at elevated ambient temperatures, and slower operational speeds, all stemming from their intrinsic operating principle, which relies on minority carriers. In contrast, the relatively less-explored NPC represents a photodetection phenomenon predominantly governed by majority carriers. In NPC, hot carriers are either trapped through interactions with self-assembled monolayers or experience reduced mobility due to phonon scattering induced by photothermal effects [3]. Previous demonstrations of NPC have involved the generation of hot carriers through the excitation of light with wavelengths either closely aligned with the surface plasmon resonance (SPR) or in proximity to the semiconductor’s band edge, resulting in inherently narrow spectral bandwidths [3]. This distinction between PPC and NPC mechanisms holds significant implications for the design and performance of photodetectors, and further exploration of the NPC phenomenon may offer promising avenues for the enhancement of the versatility and efficiency of photodetection technologies.

Semiconductor photodetectors represent a pivotal class of optoelectronic devices that are known for their ability to harness photon energy through intricate electronic processes. These devices exhibit exceptional attributes, including ultrafast response times and high responsivity, rendering them of paramount significance across a multitude of societal applications. These applications span diverse domains, such as optical communication, where photodetectors facilitate the transmission of data through light signals, as well as sensing, motion detection, missile warning, and advanced biomedical imaging systems [4]. A burgeoning area of exploration within photodetector research centers on the development of self-driven and self-powered photodetectors. This pursuit was driven by the ambition to create photodetectors capable of autonomous operation, obviating the need for external power sources. Among photodetector types, the Schottky photodiode has emerged as a notable candidate owing to its unique capacity to operate in two distinct modes [5,6]. First, the photovoltaic mode enables the Schottky photodiodes to function without bias voltage, effectively making them self-powered. In this mode, incident photons are directly converted into electrical current, offering the prospect of self-sustaining operation. Conversely, Schottky photodiodes can operate in a photoconductive mode when subjected to a reverse bias voltage. This mode empowers these devices to modulate the flow of electrical current, which is contingent upon the incident light intensity.

Schottky junction-based, self-powered photodetectors offer several compelling advantages [7,8,9]. They are distinguished by their cost-effectiveness, which stems from a straightforward and economically efficient manufacturing process. Furthermore, their device architecture is simple, facilitating seamless integration with complementary metal-oxide-semiconductor (CMOS) technologies, which are a staple in modern electronics. Notably, these photodetectors possess the innate ability to autonomously generate electrical signals, eliminating their dependence on external power supplies. Moreover, Schottky photodiodes exhibit an admirable breadth of spectral sensitivity, making them versatile instruments capable of detecting light across a broad range of wavelengths within the electromagnetic spectrum, which is highly desirable for numerous scientific and technological endeavors. As a result of strategically engineered band alignments and the distinct properties of various materials, plasmonic metal–semiconductor interfaces have gained widespread recognition and show great promise in semiconductor technologies [10,11]. They are poised to potentially replace ultra-shallow doped regions with cutting-edge CMOS technologies. Furthermore, these heterostructures serve as pivotal components within high-performance nanoelectronics devices, exerting a clear and decisive influence on the devices’ operational characteristics, including switching speed, open-circuit voltages in solar cells, and the transistor’s ON-state current and ON/OFF ratios. Conventional metallic nanoparticles, specifically gold and silver, have established themselves as prominent plasmonic materials in various practical applications, including photodetectors [12], spectroscopy [13], and novel solar energy harvesting concepts [14]. Recent advancements in the field have significantly illuminated the remarkable capacity of plasmonic nanostructures, with a primary focus on Au and Ag, to engender “hot carriers” via nonradiative decay processes [15]. This groundbreaking phenomenon of the hot carrier generation has ushered in a new frontier of possibilities across diverse applications. Notably, it has catalyzed progress in areas such as water splitting [16] and the dissociation of hydrogen [17], offering intriguing prospects for sustainable energy solutions. Furthermore, these excitations involving hot carriers have unveiled avenues for direct light-to-electricity conversion mechanisms, revolutionizing the prospects of highly efficient photovoltaic devices and photodetectors. This not only amplifies the efficiency of energy conversion but also holds the potential to redefine the landscape of renewable energy sources and the realm of optoelectronic technologies [18].

It is well known that Iridium (Ir) is a widely respected transition metal; it has attracted great attention from the scientific community as it has complex optical and electronic characteristics. For instance, one remarkable facet of iridium is its ability to introduce plasmonic behavior, an occurrence observed in the ultraviolet (UV) to near-infrared (NIR) spectrum, making it a compelling contender for a range of applications in the field of optoelectronics [1]. Moreover, the plasmonic response of iridium stems from the coordinated movement of unbound electrons at the intricate junction between the metal and dielectric substrates. Exploiting iridium’s plasmonic behavior for photodetection encompasses two fundamental mechanisms: the photoconductive and photovoltaic effects. The former entails harnessing the potent plasmonic field to enhance the photoconductivity of neighboring semiconductor materials. This is effectively achieved by integrating iridium as a plasmonic electrode alongside semiconductor counterparts such as silicon or germanium. This collaborative synergy results in a noticeable surge in photocurrent, ultimately bolstering the photodetection capabilities [2]. Furthermore, the complex interaction between the optical and electronic features of metallic iridium (Ir) is a key factor in shaping its plasmonic properties and its potential for photodetection. The electronic arrangement, marked by a d-band that is only partially filled, leads to a notable reflectivity in the visible and near-infrared (NIR) spectrum [3]. Although interband transitions in metallic iridium (Ir) control the way it interacts with light across the UV to visible spectrum, the emergence of the plasmonic behavior is rooted in different phenomena. The origin of the plasmonic response in metallic Ir can be traced to the synchronized oscillation of unbound electrons. This collective electron oscillation transpires at energy levels lower than those involved in interband transitions, resulting in a plasmonic region that encompasses wavelengths from the visible to the near-infrared (NIR) part of the spectrum. This particular spectral range has generated substantial scientific interest because of its potential applications in optoelectronics [4]. Therefore, the complex interaction between the optical and electronic intricacies in metallic iridium (Ir) presents a promising opportunity to utilize its plasmonic potential and elevate its photodetection capabilities [5]. The deliberate integration of Ir into photodetectors is expected to yield heightened stability and improved performance, highlighting its potential to make significant advancements in the field [6]. A notable illustration of its capabilities involves the use of iridium (Ir) in UV photodetectors based on localized surface plasmon enhancement within diamond materials. In this scenario, precise arrays of Ir nanoislands have been carefully crafted, and their impact on photoelectric performance has been thoroughly investigated [3,7]. Within the scope of this investigation, a comprehensive examination of a self-powered photodetector featuring an ultrathin MoO_3_/Ir/SiO_2_/Si structure was conducted. The study includes the intriguing observation of a novel NPC phenomenon that emerges after an annealing process. This phenomenon is of particular interest due to its amalgamation of aspects from both PPC and NPC within self-powered photodetector devices. Moreover, the research delves into the multifaceted role assumed by the Ir layer in the modulation of hot plasmonic effects. This dimension introduces complexity and fascinating aspects to the study, as it illuminates how the presence of the Ir layer can exert influence over and govern the behavior of these plasmonic effects. To obtain a thorough grasp of the Ir layer’s impact, an exhaustive analysis was carried out on a multitude of photodetector parameters. These parameters encompass response and recovery times, responsivity, detectivity, and external quantum efficiency (EQE). Through meticulous scrutiny of these parameters, valuable insights were garnered regarding the nuanced effects and performance enhancements that the Ir layer imparts to the self-powered photodetector.

## 2. Materials and Methods

Initially, the n-type Si substrates were cut into uniform 1 cm^2^ sections to ensure consistent sample sizes. Following this, the substrates underwent a cleaning process involving immersion in acetone, isopropyl alcohol, and deionized water within an ultrasonic bath for a duration of 20 min. Subsequently, they were dried in an oven and with the aid of nitrogen gas. To activate the substrate surfaces, an O_2_ plasma cleaner was utilized for a duration of 1 min before their introduction into the deposition process. To generate an exceedingly thin layer of iridium (Ir) measuring only a few nanometers in thickness, we employed a Leica EM ACE600 US sputter coater. Throughout this procedure, the Ir layer was deposited onto a high-vacuum film surface at a pressure of 1 × 10^3^ mbar. This was accomplished by utilizing argon plasma at a deposition pressure of 5.0 × 10^−2^ mbar and applying a current of approximately 50 mA. The width of the Ir target was 80 mm, and the layer’s thickness was monitored using a quartz thickness monitor. The sputtering rate for all the samples was 0.06 nm/s.

The MoO_3_ thin film was fabricated using an atomic layer deposition (ALD) system provided by ANRIC Technologies (AT410). An organometallic precursor material based on molybdenum, specifically bis(t-butylimido)bis(dimethylamino)molybdenum (VI), was deposited onto the substrate surface within an argon atmosphere (flowing at 20 sccm, with a purity of 99.999%) under low pressure, approximately 0.3 torrs. The quantity of material deposited onto the substrate was influenced by several variables, including the duration of the valve opening connecting the growth chamber and the precursor material chamber, the temperature within the growth chamber, and the controlled heating of the precursor material chamber (around 80 °C). To prevent condensation between the growth and material chambers, the pipes were kept at 100 °C, and the outlet manifold was maintained at 105 °C. After deposition, the gases emitted from the substrate were purged from the growth chamber at a flow rate of 10 sccm. In the subsequent stage, O_3_ (ozone) was introduced into the growth chamber via the ANRIC Technology ATO3 module. This resulted in the oxidation of the organometallic layer, leading to the creation of a monolayer MoO_3_ thin film. Finally, residual gases were evacuated to complete the cycle. Nevertheless, the influence of annealing in an air environment at 600 °C for 15 min was also investigated [19,20]. In investigating the influence of Ir, various thicknesses of 2 nm and 4 nm were deposited, with the MoO_3_ layer’s thickness held constant at 600 pulses. These depositions were carried out at 200 °C using ALD [21]. To complete the process and facilitate the creation of conductive electrodes for connectivity with the probe station, silver paste was applied.

This study employed a range of characterization techniques to investigate various aspects of the thin films, including their morphology, thickness impact, and electrical/optoelectronic properties. To observe the surface morphology, Zeiss Gemini 500 field-emission scanning electron microscopy (FESEM) was utilized. The thin film’s crystallinity and the identification of both the 2D behavior and the α-phase of the 2D α-MoO_3_ phase were examined through Raman spectroscopy. A laser source with a wavelength of 532 nm and a laser spot diameter of 2 μm was used for excitation. To assess the electrical properties of the thin film samples, the SWIN Hall8800 Hall Effect measurement system was employed to measure carrier concentration and mobility. Various parameters, including R_s_ (sheet resistance), R_ho_ (resistivity), V_H_ (Hall voltage), R_H_ (Hall coefficient), N_s_/P_s_ (sheet carrier concentration), and N/P (carrier concentration), were calculated. For electrical I-V and optoelectronics measurements, a source meter and a four-probe system were utilized. During these measurements, illumination with a 365 nm UV light lamp was applied. The study included the preparation of un-annealed MoO_3_/2 nm Ir/SiO_2_/Si, un-annealed MoO_3_/4 nm Ir/SiO_2_/Si, post-annealed MoO_3_/2 nm Ir/SiO_2_/Si, and post-annealed MoO_3_/4 nm Ir/SiO_2_/Si samples for analysis. For simplicity, these samples are represented as un-annealed Mo/2 nm Ir/SiO_2_/Si, un-annealed Mo/4 nm Ir/SiO_2_/Si, post-annealed Mo/2 nm Ir/SiO_2_/Si, and post-annealed Mo/4 nm Ir/SiO_2_/Si samples.

## 3. Results and Discussions

### 3.1. Surface Morphology 

The surface morphology of the MoO_3_ film was meticulously examined through the utilization of FESEM in both its un-annealed and post-annealed states. Figure 1 captures the essence of this analysis by showcasing the distinct surface characteristics of two specific configurations: Mo/2 nm Ir/SiO_2_/Si and Mo/4 nm Ir/SiO_2_/Si thin films. The surface morphologies of the un-annealed Mo/2 nm Ir/SiO_2_/Si and Mo/4 nm Ir/SiO_2_/Si thin films are shown in Figure 1a,c, respectively, while those of the post-annealed Mo/2 nm Ir/SiO_2_/Si and Mo/4 nm Ir/SiO_2_/Si thin films are shown in Figure 1b,d, respectively. Upon delving into these images at varying magnifications, a consistent observation emerges. Both sets of samples exhibit a profusion of closely packed particles that extend across a substantial area. This feature is maintained at both low and high magnifications. Moreover, a remarkable homogeneity is evident in the films, characterized by the presence of small grains that are uniformly dispersed. However, it is crucial to acknowledge the emergence of white larger particles in these images. These larger particles are a consequence of the sample preparation process, particularly the cutting procedure that precedes the FESEM analysis. This underscores the necessity of distinguishing between the genuine surface features and the artifacts introduced during the analytical stages. Upon a comprehensive assessment, a significant insight arises. The thin films subjected to the annealing process display a denser composition in contrast to their non-annealed counterparts. Additionally, there is a discernible trend concerning the thickness of the Ir layer. An increase in Ir thickness corresponds to an augmentation in both the size and density of the surface particles, as visibly exemplified in the images. 

Furthermore, EDS elemental analysis and mapping were carried out to gain a deeper insight into the impact of varying Ir thickness and atom diffusion. This section has been added to the Appendix A for elemental analysis. To gain a comprehensive understanding of element distribution within all the samples across the thin film, EDS layered mapping scans were conducted and generated elemental mapping distributions for the unannealed samples, as depicted in Appendix A. As anticipated, it observed well-distributed surface elements of Mo across all the samples with minimal interference and excellent integration. 

### 3.2. Raman Shift of α-MoO_3_

Raman spectroscopy stands as a potent analytical method employed to investigate the vibrational patterns of molecules. Its utility lies in the insightful data it furnishes regarding the makeup of chemicals, the arrangement of molecules, and the connections within materials. This technique reveals a distinctive pattern of peaks that represent typical MoO_3_ signatures, as showcased in Figure 2. These peaks correspond to distinct vibrational modes [22]. The most dominant of these peaks, detected at 970 cm^−1^, is assigned to the anti-symmetric elongation of terminal oxygen atoms, commonly referred to as the A_g_ mode [23,24]. This specific peak serves as confirmation for the development of the orthorhombic crystalline form of α-MoO_3_. A subsequent peak at 822 cm^−1^ indicates the symmetric elongation of bridge oxygen atoms coordinated doubly (ν(Mo-O-Mo)) [25,26,27]. The presence of this peak substantiates the material’s stable orthorhombic phase. Moreover, a relatively faint peak at 669 cm^−1^ emerges in the spectra. This particular peak is linked to the anti-asymmetric elongation of Mo-O bonds within triply coordinated oxygen atoms. When scrutinizing MoO_3_ nanopowders, a distinctive deformation mode known as δ(O-Mo-O) (B_1g_) becomes evident. This characteristic feature is represented by bands at 228 cm^−1^ in MoO_3_ nanopowders (B_2g_) and at 301 cm^−1^ in MoO_3_ nanofilms (B_3g_). These bands are associated with the δ(O-Mo-O) wagging motion [22]. When comparing MoO_3_ nanofilms with their commercial MoO_3_ nanopowder counterparts, a noticeable phenomenon emerges: the peaks exhibited by the nanofilms appear broader and exhibit slight shifts. This alteration in peak characteristics can be attributed to the confinement of phonons due to the diminished grain size and the dimensional attributes of the ultra-thin nanofilms [28,29]. An intriguing observation pertains to the peak at 822 cm^−1^, which essentially signifies the presence of a layered 2D α-MoO_3_ structure. These findings align with those of previous reports [27,30,31,32]. The peaks at 1298, 1317, and 1442 cm^−1^ are closely associated with Si, which is likely indicative of its influence within the samples. A key factor here is the integration of an Ir layer, which functions as a plasmonic metal layer. This layer possesses the capability to enhance light reflection compared to that of the oxide layer of MoO_3_. This heightened light interaction significantly influences the Raman peaks. Consequently, a discernible pattern emerges: as the thickness of the Ir layer is increased, there is a corresponding augmentation in the intensity of the Raman peaks. This outcome implies a distinct correlation between the Ir layer’s thickness and the amplified crystalline behavior of the MoO_3_ layer. This relationship is particularly evident when comparing higher Ir thicknesses to lower ones.

### 3.3. Electrical Properties

The electrical characteristics of the thin films made from MoO_3_ and Ir layers on Si substrates, both before and after undergoing an annealing process, were carefully examined. These findings are visually presented in Figure 3. The analysis outcomes reveal some interesting insights into how the thickness of the Ir layer plays a crucial role in the electrical performance of these films. Upon close observation, it becomes evident that altering the thickness of the Ir layer has a noticeable impact on the electrical properties of the films. However, the effects are particularly pronounced after the annealing process has been carried out. This annealing procedure involves subjecting the films to controlled heating, which appears to significantly influence their behavior.

One intriguing observation is that the annealing treatment leads to an augmentation in both the dark current (the current flowing through the film in the absence of any external illumination) and the photocurrent (the current generated when the film is exposed to light) for both the samples with 2 nm and 4 nm thick Ir layers. Furthermore, a distinctive differentiation emerges when comparing the two samples. The film of Mo/4 nm Ir/SiO_2_/Si exhibits what is termed ohmic behavior under dark conditions. This essentially means that the electrical behavior of this film demonstrates a linear relationship between the voltage and the current, indicating a straightforward conduction mechanism. On the other hand, the situation is notably different for the film where the Ir layer is a mere 2 nm thick. This film showcases what is known as Schottky behavior when subjected to UV illumination. This type of behavior implies a more complex conduction mechanism influenced by the interaction between the metal and the semiconductor.

### 3.4. Zero Bias ON–OFF Dynamic Behavior

The performance of photodetectors and optoelectronic devices is depicted through their ON–OFF curve. This curve illustrates their response over time when exposed to light (ON) or when the light source is turned off (OFF state) under a specific bias voltage. It is crucial to repeat this process multiple times to grasp the devices’ overall behavior over time. Applying a bias voltage is known to enhance photocarrier generation and, subsequently, photodetection capacity. However, there is a growing interest in self-powered optoelectronic devices, like zero-voltage operating devices. These can be activated under specific conditions, as seen in p–n junction photodetectors [33,34,35], Schottky junction photodetectors [5,9,36], and photoelectrochemical-type (PEC-type) photodetectors [37,38,39]. Figure 4 illustrates the ON–OFF photoresponse for both un-annealed and post-annealed conditions. In the case of the non-post-annealed MoO_3_ layer on Ir/SiO_2_/Si, the MoO_3_ is deposited via ALD at 200 °C [21]. In contrast, samples subjected to annealing treatments undergo a rapid annealing process at 600 °C in air, as optimized earlier [19]. Under un-annealed conditions, the Mo/4 nm Ir/SiO_2_/Si sample exhibits a higher ON–OFF ratio than that of the Mo/2 nm Ir/SiO_2_/Si sample. This observation aligns with the anticipated behavior based on IV electrical curves, suggesting greater conductivity in the Mo/4 nm Ir/Si sample. Notably, the ON–OFF results reveal a minor decay in the ON state for Mo/4 nm Ir/SiO_2_/Si, which is absent in the Mo/2 nm Ir/SiO_2_/Si case, indicating higher dynamic stability in the latter. These findings suggest positive photoconductivity in the un-annealed samples, along with low dark current in both the un-annealed and post-annealed scenarios. Conversely, the samples subjected to annealing exhibit negative photoconductivity behavior at a 0 V bias voltage. The ON–OFF ratio is higher for the Mo/2 nm Ir/SiO_2_/Si sample compared to that of the Mo/4 nm Ir/SiO_2_/Si. Both samples demonstrate consistent signal dynamics over time.

To gain deeper insights into this behavior, conducting electrical Hall effect measurements on these samples becomes crucial, as shown below. This approach allows us to comprehend the impact of annealing on factors such as carrier density and sheet resistances. By examining these parameters, a more comprehensive understanding of the observed behavior can be achieved.

### 3.5. Effect of Carrier Concentrations on the Positive and Negative Photoconductivity

The electrical properties of both the un-annealed and the post-annealed samples could be understood firstly via the Hall effect measurements. This measurement was conducted at room temperature and under dark conditions with a constant magnetic field of B = 6800 G, as recorded in Table 1. The parameters under investigation encompass R_s_ (sheet resistance), R_ho_ (resistivity), V_H_ (Hall voltage), R_H_ (Hall coefficient), N_s_/P_s_ (sheet carrier concentration), N/P (carrier concentration), and Mob (mobility). These metrics collectively offer valuable insights into the electrical characteristics and behavior of the samples; in particular, they shed light on the effects of annealing on carrier density, sheet resistance, and other fundamental properties. At the outset, all the samples exhibited characteristics indicative of p-type semiconductor behavior. Notably, the Rs value demonstrates a decline with increased air thickness and the application of annealing treatment. In the case of the un-annealed Mo/2 nm Ir/SiO_2_/Si sample, the N_s_ value registers at 5.76 × 10^11^/cm², which subsequently increases to 6.74 × 10^12^/cm² after the annealing treatment. A parallel trend is evident in the N values, which ascend from 7.96 × 10^17^/cm³ to 9.31 × 10^18^/cm³. This underscores how the annealing treatment bolsters the conductivity of the Mo/2 nm Ir/SiO_2_/Si sample and elucidates the higher ON–OFF ratio following the annealing process. Conversely, the un-annealed Mo/4 nm Ir/SiO_2_/Si sample records an Ns value of 2.99 × 10^10^/cm², diminishing to 2.80 × 10^10^/cm² following the annealing treatment. Similar behavior is observed in the N values, which decline from 3.18 × 10^16^ to 2.99 × 10^16^/cm³. This decrease in N_s_ and N values after annealing treatment chiefly contributes to the diminished ON–OFF curves observed in the Mo/4 nm Ir/SiO_2_/Si sample after undergoing the annealing treatment.

### 3.6. Origin of Negative Photoconductivity

The initial observation of the NPC effect might be attributed to the presence of adsorbed H_2_O molecules on the surface, acquired during the annealing treatment process in air. The significant influence of absorbed and subsequently photodesorbed H_2_O molecules on photoelectric characteristics is well documented; it leads to the induction of the NPC effect in nanoscale materials [40,41]. The significance of surface effects stemming from oxygen adsorption and photodesorption has been acknowledged as pivotal in the modulation of the photoresponse of semiconductor nanostructures. This holds for various cases, including single-walled carbon nanotubes [42], ZnO nanowires [43,44], the CH_3_NH_3_PbBr_3_–ZnO heterostructure [45] and CdS nanoribbons [46,47]. A similar underlying mechanism could potentially underlie negative photoresponses. Broadly, the water molecules present on a semiconductor’s surface act as electron donors [48,49,50], amplifying the electron concentration within the material. When in dark ambient conditions, oxygen molecules tend to adsorb onto the surface of p-type ZnSe nanowires [51]. Subsequently, they become ionized through electron capture from the nanowire due to its pronounced electronegativity. A similar situation was observed for ZnSe nanowires [52].

### 3.7. Transient Behavior Analysis

It is imperative to assess the temporal stability characteristics of the sensor. Both the un-annealed and post-annealed samples, featuring Ir layers with thicknesses of 2 nm and 4 nm, exhibit distinct responses under conditions of darkness and UV illumination, as depicted in Figure 5. Under dark conditions, both the 2 nm and 4 nm Ir samples exhibit an increase in current for the post-annealed specimens. As illustrated in Table 1, this augmented current is attributed to the improved crystallinity, which facilitates the mobility of charge carriers, based on the Hall effect measurements. Conversely, when exposed to illumination, the photocurrent begins to decline, resulting in negative photoconductivity. In the case of the 2 nm Ir sample, annealing treatment consistently yields a negative photocurrent characterized by high temporal stability. However, for the 4 nm Ir sample, stability diminishes along with the reduced photocurrent compared to the 2 nm Ir sample. These findings corroborate the earlier observations about the ON–OFF behavior.

Further confirmation was pursued through a comprehensive comparison between the responses of the un-annealed and post-annealed specimens under varying dark and UV illuminating conditions, as exemplified in Figure 6. In a broad overview, the outcomes indicate a distinct photocurrent behavior, with the un-annealed samples exhibiting a positive photocurrent response and the post-annealed samples manifesting a negative photoresponse upon exposure to illumination. This divergence in photocurrent behavior suggests a notable distinction in the charge carrier dynamics between these two sample categories. Specifically, it is noteworthy that the 2 nm Ir thick samples consistently display higher levels of current when contrasted with their 4 nm counterparts, regardless of whether they have undergone the annealing process. This observation underscores the influential role of both the annealing and Ir layer thickness in modulating the photocurrent characteristics of the sensor and has crucial implications for its performance and utility.

### 3.8. Response and Recovery Time Analysis

The assessment of the response and recovery times, with a particular focus on the influence of annealing processes, holds pivotal significance in the comprehension of the overall performance characteristics of the self-powered photodetector. The corresponding results are graphically presented in Figure 7. In a general context, it is evident that the response time exhibits a shorter duration for the 2 nm Ir thick sample as compared to its 4 nm Ir counterpart, under both the un-annealed and post-annealed conditions. This swifter response time can be attributed to the enhanced carrier mobility properties inherent to the 2 nm Ir samples, which are inherently linked to the electrical characteristics of these specimens. Conversely, when examining the recovery time, a slight delay is observed for the 2 nm Ir sample in comparison to the 4 nm Ir sample. Nevertheless, it is worth noting that both sample categories demonstrate remarkably rapid response and recovery times, typically falling within the range of 0.1 to 0.6 milliseconds. Furthermore, the 2 nm Ir sample distinguishes itself by showcasing the shortest response time of just 0.1 milliseconds. These findings underscore the critical role played by both the annealing processes and the thickness of the Ir layer in shaping the response and recovery dynamics of the self-powered photodetector and thereby offer valuable insights into its operational capabilities and potential applications.

### 3.9. Assessment of Self-Powered Photodetector Parameters

In addition to scrutinizing the response and recovery times, it is imperative to delve into several other critical parameters that serve as essential benchmarks for evaluating the overall performance of the photodetector. Specifically, we turn our attention to photocurrent gain, responsivity, EQE, and detectivity, which are assessed under both un-annealed and post-annealed conditions for Mo/2 nm Ir/SiO_2_/Si and Mo/4 nm Ir/SiO_2_/Si configurations, as represented graphically in Figure 8. It is worth noting that these parameters have been extensively explored and discussed in numerous research articles [53,54]. As anticipated, the 2 nm thick Ir sample demonstrates superior performance characteristics following the annealing treatment. In this regard, the sample transforms into exceedingly thin plasmonic films characterized by elevated carrier concentrations and accelerated charge mobility. This transformation leads to notable enhancements across various photodetector parameters. For instance, the detectivity metric exhibits marked improvements under annealing treatment, reflecting the heightened sensitivity and detection capabilities of the photodetector. Conversely, the inverse scenario unfolds for the 4 nm thick Ir samples; thus, it is in alignment with the previously observed trends in optoelectronic measurements. Annealing treatment adversely affects the photodetector’s performance, resulting in diminished photodetection capabilities. These findings collectively underscore the pivotal role played by annealing processes and the thickness of the Ir layer in influencing the photodetector’s key performance parameters and thereby provide valuable insights into its operational efficacy and potential applications.

## 4. Conclusions

This study presents an investigation into a self-powered photodetector device harnessing the plasmonic effects of ultrathin Ir (Iridium) films. Specifically, MoO_3_/Ir/SiO_2_/Si structures with Ir thicknesses of 2 nm and 4 nm were meticulously fabricated using sputtering and atomic layer deposition systems, which are primarily intended for ultraviolet photodetection applications. Notably, the impact of annealing treatment on these structures is a key focal point, revealing a novel observation of negative photoconductivity within the ultrathin MoO_3_/Ir/SiO_2_/Si films. The examination of surface morphology, conducted through field-emission scanning electron microscopy, provides invaluable insights into the dynamic transformations induced by annealing in the MoO_3_ film. These visual representations encapsulate the evolving characteristics related to density, particle size, and homogeneity, with the added dimension of varying Ir layer thicknesses contributing to this dynamic. Crucially, the study of these MoO_3_/Ir/SiO_2_/Si thin films, both before and after annealing, highlights the substantial influence exerted by the thickness of the Ir layer on their electrical behavior. Furthermore, the annealing treatment accentuates specific electrical traits, thereby shedding light on the intricate interplay between material properties and the external annealing conditions they undergo. Notably, the response time of these devices was remarkably swift, measuring 0.1 milliseconds under a bias voltage of 0 V. This characteristic underscores the high-speed performance achievable with these self-powered photodetectors.

## Figures and Tables

**Figure 1 materials-16-06756-f001:**
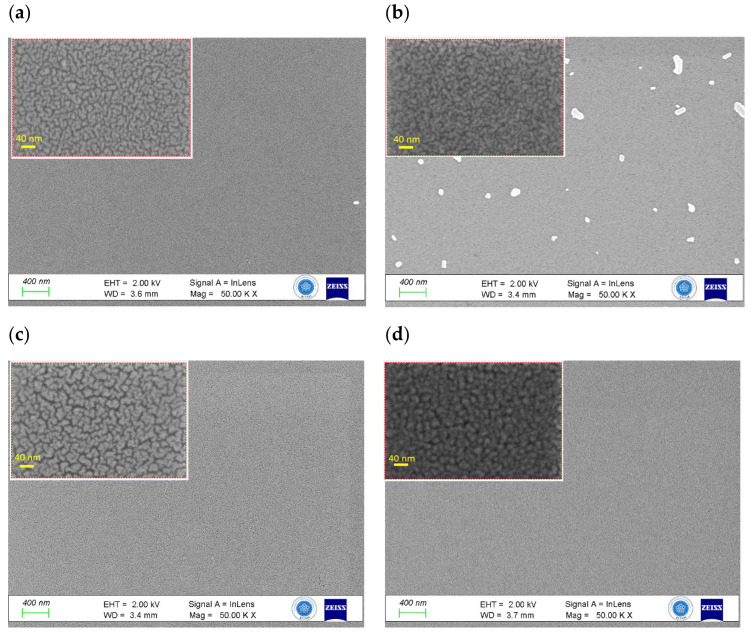
FESEM images comparing un-annealed (**a**,**c**) and post-annealed (**b**,**d**) samples of a MoO_3_/Ir/SiO_2_/Si thin film with Ir thicknesses of 2 nm and 4 nm, respectively. The main images were captured at a magnification of 50,000× with a 400 nm scale bar. Insets showcase higher magnification views at 600,000× with a scale bar of 40 nm.

**Figure 2 materials-16-06756-f002:**
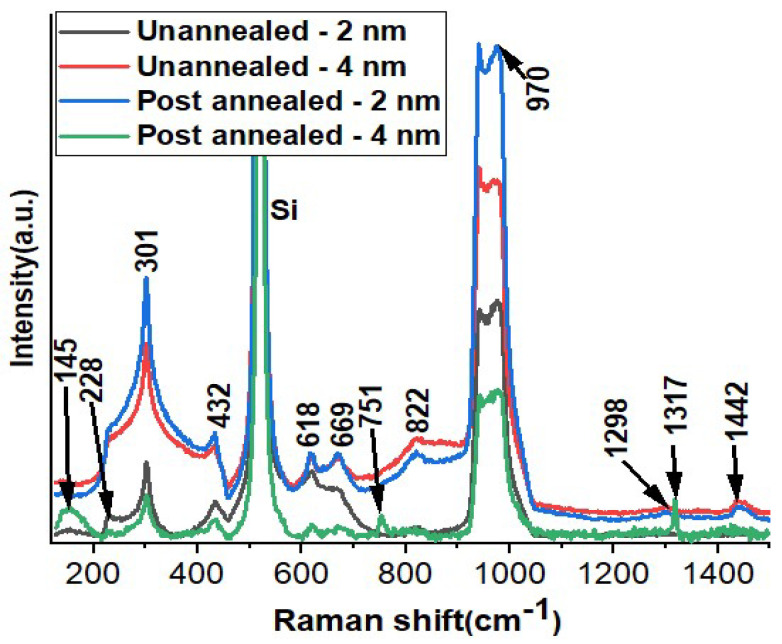
Raman shift analysis of un-annealed and post-annealed effects on Mo/2 nm Ir/SiO_2_/Si and Mo/4 nm Ir/SiO_2_/Si thin films.

**Figure 3 materials-16-06756-f003:**
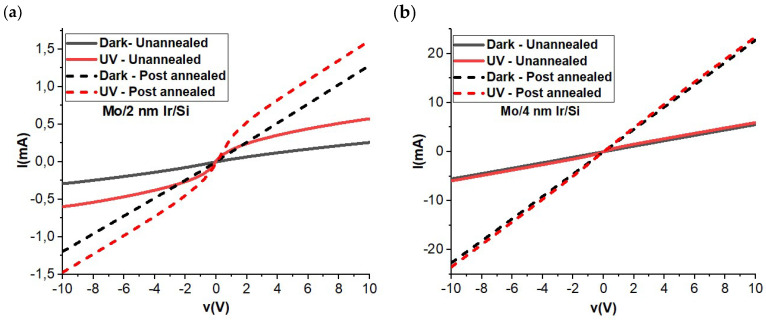
Electrical IV curves of Mo/2 nm Ir/SiO_2_/Si (**a**) and Mo/4 nm Ir/SiO_2_/Si (**b**) with un-annealing and annealing conditions.

**Figure 4 materials-16-06756-f004:**
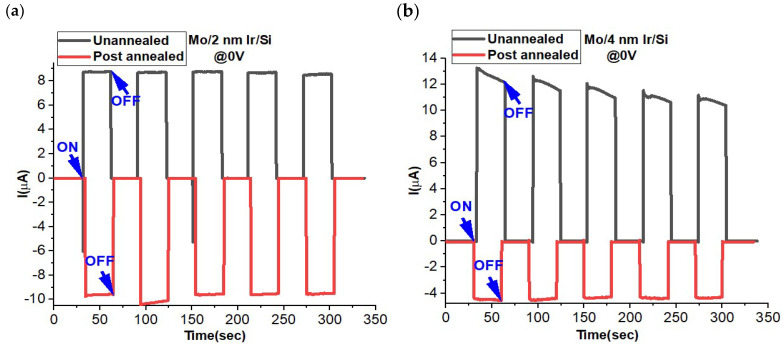
ON–OFF performance Mo/2 nm Ir/SiO_2_/Si (**a**) and Mo/4 nm Ir/SiO_2_/Si (**b**) under un-annealing and annealing conditions at 0 V bias.

**Figure 5 materials-16-06756-f005:**
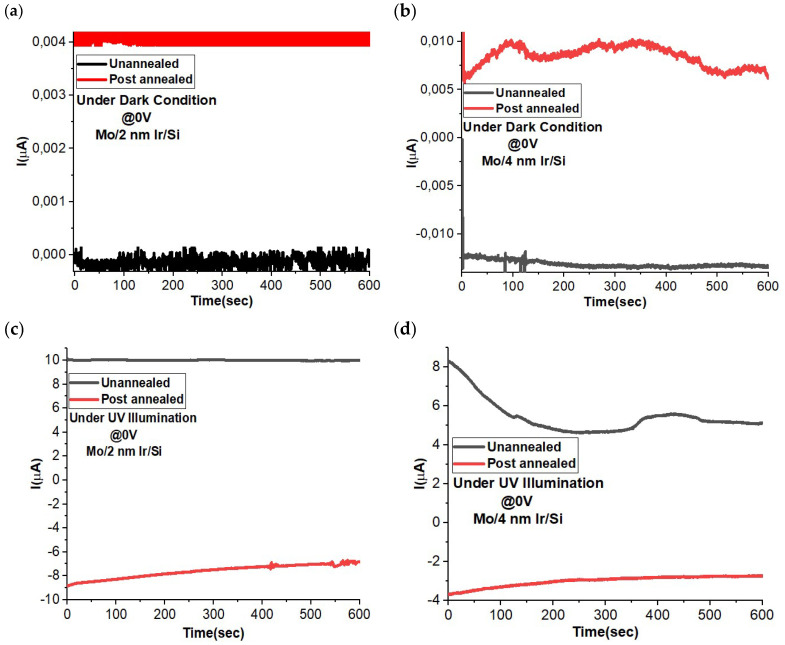
Current–time transient behavior under dark and UV illumination of Mo/2 nm Ir/SiO_2_/Si (**a**,**c**) and Mo/4 nm Ir/SiO_2_/Si (**b**,**d**) for un-annealing and annealing treatment. The measurements were collected at 0 V bias voltage for self-power photodetector measurements.

**Figure 6 materials-16-06756-f006:**
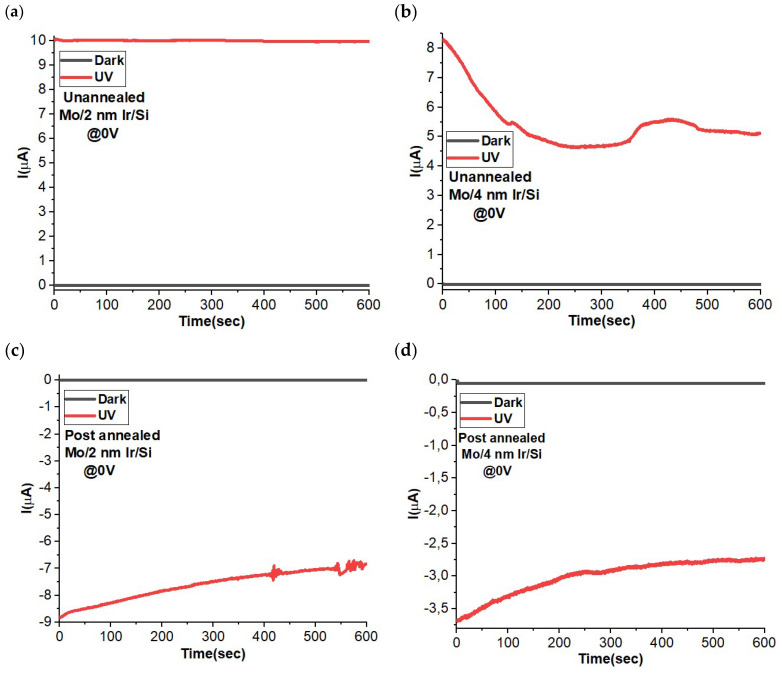
Current–time transient behavior under dark and UV illumination of un-annealed Mo/2 nm Ir/SiO_2_/Si (**a**), un-annealed Mo/4 nm Ir/SiO_2_/Si (**b**), annealing treatment Mo/2 nm Ir/SiO_2_/Si (**c**), and annealing treatment Mo/4 nm Ir/SiO_2_/Si (**d**). The measurements were collected at 0 V bias voltage for self-power photodetector measurements.

**Figure 7 materials-16-06756-f007:**
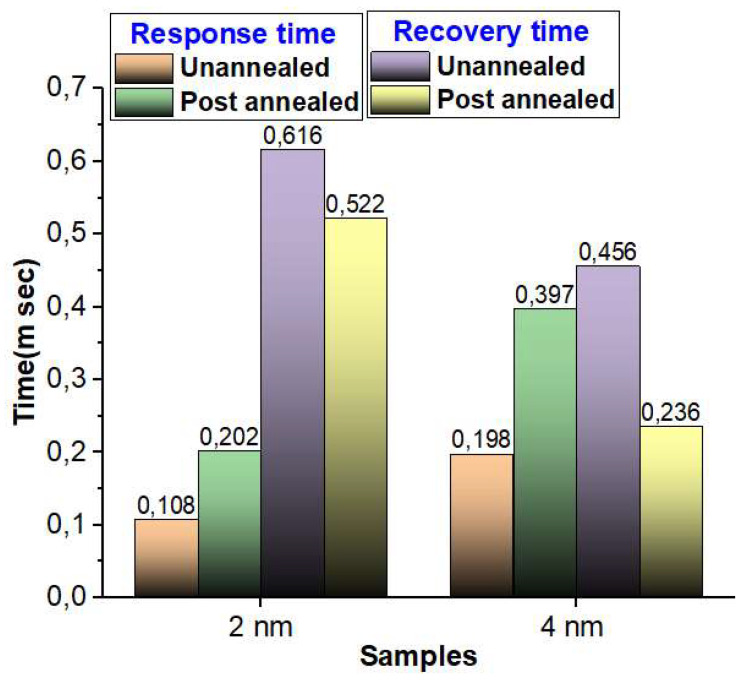
Response and recovery time of un-annealed and post-annealed MoO_3_/Ir/SiO_2_/Si thin films of 2 nm and 4 nm Ir layer.

**Figure 8 materials-16-06756-f008:**
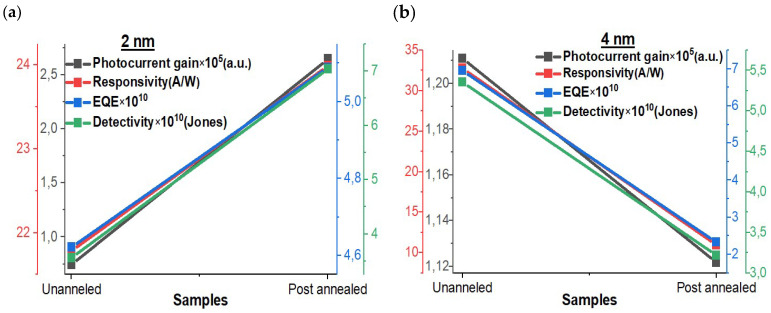
Photocurrent gain, responsivity, EQE, and detectivity under un-annealed and post-annealed conditions of Mo/2 nm Ir/SiO_2_/Si (**a**) and Mo/4 nm Ir/SiO_2_/Si (**b**).

**Table 1 materials-16-06756-t001:** Hall effect measurement results for un-annealed and post-annealed Mo/2 nm Ir/SiO_2_/Si and Mo/4 nm Ir/SiO_2_/Si samples.

Condition	Sample	Rs (Ω/sq)	R_ho_ (/Ω-cm)	V_H_ (V)	R_H_ (m^3^/C)	Type	N_s_ (/cm^2^)	N (/cm^2^)	Mobility (cm^2^/Vs)
Un-annealed	Mo/2 nm Ir/SiO_2_/Si	1.71 × 10^5^	1.24 × 10^−1^	2.34 × 10^−4^	7.84 × 10^−6^	P	5.76 × 10^11^	7.96 × 10^17^	6.31 × 10^1^
	Mo/4 nm Ir/SiO_2_/Si	1.16 × 10^5^	1.09 × 10^−1^	1.51 × 10^−2^	1.96 × 10^−4^	p	2.99 × 10^10^	3.18 × 10^16^	1.80 × 10^3^
Post-annealed	Mo/2 nm Ir/SiO_2_/Si	5.34 × 10^3^	3.87 × 10^−3^	6.68 × 10^−5^	6.71 × 10^−7^	P	6.74 × 10^12^	9.31 × 10^18^	1.73 × 10^2^
	Mo/4 nm Ir/SiO_2_/Si	2.86 × 10^3^	2.68 × 10^−3^	4.82 × 10^−3^	2.09 × 10^−4^	P	2.80 × 10^10^	2.99 × 10^16^	7.80 × 10^4^

## Data Availability

The data presented in this study are available on request from the corresponding authors.

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
