# Peer review of "Negative Photoconductivity in 2D α-MoO_3_/Ir Self-Powered Photodetector: Impact of Post-Annealing"

_materials, 2023, doi:10.3390/ma16206756_

Round 1
Reviewer 1 Report
This paper presents the observation of negative photoconductivity found from MoO3 coated with thin (2 or 4nm) Ir contact layer. It concludes that diffusion of Ir during annealing treatment makes MoO3 to exhibit negative photoconductivity especially when Ir is 2nm thick. The paper is very well written except for a few casual errors (explain English section) and the work is very well conducted. However, following issues may need to be addressed in order to enjoy deserving recognition of the accomplishment made in this work:
1) description of measurement method: the title and a good portion of introduction produces impression that the electrical measurement presented in this paper would be related to the junction. However, it seems that non of properties reported in this paper is related to the junction. It is the property of MoO3 itself (with Ir coating). This needs to be clearly stated in order to avoid confusion;
2) in relation 1), it also needs to be stated somewhere that the negative current in Fig.4 is not real but is taken to show the reduced photoconductivity. If the negative current is true in Fig.4, then it means major carrier is changed by "anneal" and UV. Then, Ir is acting as a dopant creating an opposite carrier, which shouldn't be the case.
3) The negative photoconductivity is related to diffusion of 2nm Ir into MoO3. If it is sorely a diffusion (and increase the phonon scattering), then why doesn't same effect happen with 4nm Ir? This is the part that is not very well articulated.
English is good except for a few places of casual errors. For example, phrases like "increase xxx after post-annealing treatment" appears in a few places (in similar nature). The correct wording should be "after annealing treatment". Careful proof-reading is recommended to remove casual errors like this.
Author Response
The authors would like to extend their heartfelt appreciation for your dedicated time and contributions. Further attached, you will find comprehensive responses addressing all of your inquiries.

Reviewer 2 Report
Review for Negative Photoconductivity in 2D α-MoO3/Ir Schottky Junction Self-Powered Photodetector: Impact of Post-Annealing on Surface Diffusion of Iridium Atoms
In the manuscript, the authors introduced the concept of negative photoconductivity and Schottky junction-based self-powered photodetectors to illustrate the MoO3/Ir/Si film with and without thermal annealing. The manuscript fits the journal topic well. However, there are several major questions regarding the data and story before moving further.
1. In the introduction, Line 25, "This suggests...". The carrier concentration change may not directly lead to the following explanation. The analysis of carrier concentration change could suggest the Ir diffusion but could be something else. Only Raman is not strong enough to claim. Ir diffusion may need intrinsic characterization methods, such as FIB-SEM or cross-section SEM/EPS.
2. Si wafer likely has oxidation layers, even before the O2 plasma cleaning. It is suggested that use MoO3/Ir/SiO2/Si instead of MoO3/Ir/Si since the SiO2 layer should be thicker than the Ir layer.
3. With the second point, in section 3.2, please include the Si/SiO2 wafer Raman spectrum as a reference. Typically, the peaks at (970 cm-1) shown in Figure 2 can be observed in the Si/SiO2 Raman spectra, which are the 2TO peaks. Some claims in the Raman spectra may not be valid.
Author Response

(The authors gave the same response as above.)
